# Association between Nafamostat Mesylate and In-Hospital Mortality in Patients with Coronavirus Disease 2019: A Multicenter Observational Study

**DOI:** 10.3390/jcm11010116

**Published:** 2021-12-26

**Authors:** Ryota Inokuchi, Toshiki Kuno, Jun Komiyama, Kazuaki Uda, Yoshihisa Miyamoto, Yuta Taniguchi, Toshikazu Abe, Miho Ishimaru, Motohiko Adomi, Nanako Tamiya, Masao Iwagami

**Affiliations:** 1Department of Health Services Research, Faculty of Medicine, University of Tsukuba, Tsukuba 305-8575, Japan; inokuchir-icu@md.tsukuba.ac.jp (R.I.); jun.komi33@gmail.com (J.K.); abetoshi111@gmail.com (T.A.); ishimaru.miho.kf@u.tsukuba.ac.jp (M.I.); adomimotohiko@gmail.com (M.A.); ntamiya@md.tsukuba.ac.jp (N.T.); 2Montefiore Medical Center, Division of Cardiology, Albert Einstein College of Medicine, New York, NY 10461, USA; tkuno@montefiore.org; 3Graduate School of Comprehensive Human Sciences, University of Tsukuba, Tsukuba 305-8575, Japan; taniguchi.yuta.ma@alumni.tsukuba.ac.jp; 4Health Services Research and Development Center, University of Tsukuba, Tsukuba 305-8575, Japan; udakazuaki-tky@umin.ac.jp; 5National Cancer Center, Institute for Cancer Control, Tokyo 104-0045, Japan; ymiyamoto70@gmail.com; 6Department of Emergency and Critical Care Medicine, Tsukuba Memorial Hospital, Tsukuba 305-8575, Japan

**Keywords:** coronavirus disease 2019, in-hospital mortality, nafamostat mesylate

## Abstract

Nafamostat mesylate may be effective against coronavirus disease 2019 (COVID-19). However, it is not known whether its use is associated with reduced in-hospital mortality in clinical practice. We conducted a retrospective observational study to evaluate the effect of nafamostat mesylate in patients with COVID-19 using the Medical Data Vision Co. Ltd. hospital-based database in Japan. We compared patients with COVID-19 who were (*n* = 121) and were not (*n* = 15,738) administered nafamostat mesylate within 2 days of admission between January and December 2020. We conducted a 1:4 propensity score matching with multiple imputations for smoking status and body mass index and combined the 20 imputed propensity score-matched datasets to obtain the adjusted odds ratio for in-hospital mortality. Crude in-hospital mortality was 13.2% (16/121) and 5.0% (790/15,738), respectively. In the propensity score-matched analysis with multiple imputations, the adjusted odds ratio (use vs. no use of nafamostat mesylate) for in-hospital mortality was 1.27 (95% confidence interval: 0.61–2.64; *p* = 0.52). Sensitivity analyses showed similar results. The results of this retrospective observational study did not support an association between nafamostat mesylate and improved in-hospital outcomes in patients with COVID-19, although further studies with larger sample sizes are warranted to assess the generalizability of our findings.

## 1. Introduction

The coronavirus disease 2019 (COVID-19) pandemic is caused by severe acute respiratory syndrome coronavirus 2 (SARS-CoV-2) infection. Although worldwide vaccination against SARS-CoV-2 is underway [1], it is imperative that researchers quickly identify suitable drugs for repurposing as COVID-19-specific therapies.

Nafamostat mesylate (NM) is a serine proteinase inhibitor that has been used in Japan for over 30 years to treat disseminated intravascular coagulation and pancreatitis [2]. Experimental studies have shown that SARS-CoV-2 exerts an effect on human angiotensin-converting enzyme 2, enabling it to invade cells and establish infection [3]. In 293FT cells (derived from human fetal kidneys) that ectopically express angiotensin-converting enzyme 2, NM prevents SARS-CoV-2 spike protein-initiated fusion by inhibiting protease [4]. Thus, it can be hypothesized that NM is effective against COVID-19.

Recent previous basic experimental [5] and case reports or series [2,6,7,8] have suggested the potential effectiveness of NM against COVID-19. However, to our knowledge, neither clinical trials nor observational studies, except for a recent small phase 2 open-label, randomized controlled trial (RCT) [9] have demonstrated an association between NM and reduced mortality. Therefore, we aimed to evaluate the effect of NM in patients with COVID-19 using a large-scale in-patient database in Japan.

## 2. Materials and Methods

### 2.1. Data Source

We conducted a retrospective observational cohort study using the Medical Data Vision (MDV) Co., Ltd. (Tokyo, Japan) hospital-based database. The MDV is a private database that has been used to compile data on healthcare resource consumption at participating hospitals since 2008, based on information from the Japanese Diagnosis Procedure Combination (DPC) fixed-payment reimbursement system. The MDV database currently covers over 350 facilities in Japan, accounting for more than 20% of acute-care hospitals that use the DPC, and includes data on 32 million patients of all ages. The MDV database contains the following information: age; sex; height; body weight; admission and discharge dates; discharge status; level of consciousness; comorbidities on admission; smoking status; primary admission diagnosis according to the International Classification of Diseases, Tenth Revision (ICD-10) codes; and daily records of drugs, blood products, and procedures (including noninvasive positive-pressure ventilation [NPPV], mechanical ventilation, renal replacement therapy [RRT], and extracorporeal membrane oxygenation [ECMO]). A previous validation study of DPC data [10] suggested high sensitivity and specificity for procedural records, but high or moderate sensitivity for most diagnoses.

### 2.2. Study Participants and Exposure Variable

We included patients ≥18 years of age who met the following criteria: (1) admission diagnosis of COVID-19 in accordance with ICD-10 code U071 and (2) discharged between 1 January 2020 and 31 December 2020. We excluded patients discharged on the day of admission to avoid immortal time bias [11], meaning that patients needed to survive to receive NM. Additionally, only initial hospitalizations were included for each patient; readmissions were excluded from the study. We compared the outcomes of patients who were administered NM within 2 days after admission (NM group) to the outcomes of those who were not administered NM (control group).

### 2.3. Outcome and Covariates

The primary outcome was in-hospital mortality. Covariates included age; sex; prior diagnoses of diabetes mellitus, cardiac disease, cerebral infarction, liver disease, chronic lung disease, cancer, and chronic kidney disease; Charlson comorbidity index [12]; body mass index (BMI); smoking status; level of consciousness at the time of admission; use of antiplatelet or anticoagulant drugs (including vitamin K antagonists and direct oral anticoagulants) at the time of admission; transfer from another hospital; and admission to the intensive care unit (ICU) (ICD-10 codes are shown in Appendix A). For the purpose of analysis, we categorized levels of consciousness, according to the Japan Coma Scale (JCS), as follows: alert, JCS 0; awake without stimulation, JCS 1–3; arousable with stimulation, JCS 10–30; and unarousable, JCS 100–300. The JCS is widely used in Japan and correlates well with the Glasgow Coma Scale [13]. In addition, we identified patients who received the following treatments within 2 days of admission: NPPV, mechanical ventilation, RRT, ECMO, transfusion (including red blood cell concentrates, fresh frozen plasma, and platelet concentrates), vasoactive agents (including norepinephrine, dobutamine, and vasopressin), intravenous antibiotics, anticoagulants (including heparin and daluteparin), and steroids (including dexamethasone and other oral and intravenous steroids).

### 2.4. Statistical Analysis

Baseline characteristics that were measured as continuous or categorical variables were summarized. Categorical data are expressed as percentages. Normally and non-normally distributed variables are expressed as mean (standard deviation) and median (interquartile range), respectively. The chi-square test was used to compare categorical data, except when the expected cell counts were five or fewer, in which case Fisher’s exact test was used. Continuous variables were compared using Welch’s t-test or the Mann–Whitney U test, depending on the distribution of the data.

For our study population, we grouped patients based on the presence or absence of each diagnostic or procedural code, assuming that patients without a code did not have the corresponding condition. Therefore, no covariate data were missing, except for smoking status and BMI. There were missing values for BMI and smoking status on admission, which may have affected the results. Therefore, before propensity score matching, we replaced each missing value with a set of substituted plausible values using a multistep approach. First, we performed multiple imputations to account for missing data on smoking status and BMI [14]. We replaced each missing value with a set of substituted plausible values by creating 20 filled-in complete datasets, with 10 iterations per dataset, using the multiple imputations by chained equations method [15]. The following covariates were used in the imputation model: age, sex, Charlson comorbidity index, level of consciousness, use of antiplatelet or anticoagulant drugs, ICU admission, transfer from another hospital, comorbidities, treatments performed within 2 days of admission, and in-hospital mortality. Second, to estimate the propensity score, we fitted a logistic regression model for NM use as a function of the patient and hospital factors (i.e., age, sex, comorbidities, BMI, smoking status, level of consciousness, use of antiplatelet or anticoagulant drugs, ICU admission, transfer from another hospital, and the number of beds) and treatments performed within 2 days of admission (i.e., NPPV, mechanical ventilation, RRT, ECMO, transfusion [including red blood cell concentrates, fresh frozen plasma, and platelet concentrates], vasoactive agents [including norepinephrine, dobutamine, and vasopressin], intravenous antibiotics, and anticoagulants [including unfractionated or low-molecular-weight heparin, dexamethasone, and other steroids]). We selected these covariates based on clinical importance to calculate the propensity score [16]. Third, propensity score matching was performed for each imputed dataset using 1:4 nearest-neighbor matching based on the estimated propensity score of each patient. A match occurred when a patient in the NM-user group had an estimated propensity score within 0.2 standard deviations of that of a patient in the non-user group [17,18]. Balance was determined using standardized mean differences. To assess balance after matching, we calculated the standardized mean difference for each dataset and considered values <0.1 to be acceptable [19]. Fourth, effect estimates were determined and the results were pooled using Rubin’s rules [20]. Effect estimates are presented as odds ratios (ORs) for binary outcome data with corresponding 95% confidence intervals. Imputed and matched data are presented as pooled data.

A two-sided *p* < 0.05 was considered statistically significant. We conducted sensitivity analyses to test the robustness of our findings. First, we excluded patients who had undergone intermittent or continuous RRT from the study population, as NM is sometimes used for this purpose in Japan [21]. Second, we analyzed the outcomes using complete cases for BMI and smoking status.

Data were analyzed using JMP 15.1 (SAS Institute Inc., Cary, NC, USA), Stata MP15.1 (STATA Corp., College Station, TX, USA), and R version 4.1.1 (R Foundation for Statistical Computing, Vienna, Austria) with the mice, MatchThem, and cobalt survey packages.

## 3. Results

We included 15,859 patients after applying the inclusion and exclusion criteria (Figure 1). Of these, 2378 (15.1%) and 2784 (17.6%) patients from the NM and non-NM groups had missing data on BMI and smoking status, respectively. 

Table 1 and Table 2 show the baseline characteristics of the unmatched groups after multiple imputations. The unmatched data indicated that patients administered NM were more likely to be older males; have the comorbidities of diabetes mellitus and chronic kidney disease; have a high BMI; be in a comatose state; use antiplatelet drugs; have been transferred from another hospital; be admitted to a large-scale hospital or the ICU; be administered antibiotics, heparin, vasopressors, steroids, or transfusions; and require mechanical ventilation, RRT, or ECMO within 2 days of admission. Propensity score matching produced balanced, well-matched treatment groups for each set of imputed and pooled data (Figure 2 and Appendix A).

In the unmatched cohort, in-hospital mortality was 5.1% (806/15,859), and that of patients with or without NM was 13.2% (16/121) and 5.0% (790/15,738), respectively (*p* < 0.001). After propensity score matching, in-hospital mortality did not differ significantly between patients with or without NM (odds ratio, 1.27; 95% confidence interval: 0.61–2.64; *p* = 0.52) (Table 3). The results of sensitivity analyses (1) excluding patients who had undergone intermittent or continuous RRT and (2) using complete cases for BMI and smoking status, were similar to those of our main analysis, which did not suggest that NM had a statistically significant survival benefit among patients with COVID-19 (Table 3).

## 4. Discussion

In the present observational study, which used a large-scale in-patient database, we compared in-hospital mortality between patients with COVID-19 who had and had not been administered NM. The results of propensity score matching after multiple imputations indicated no statistically significant differences in the outcomes of the two groups. Several subsequent sensitivity analyses also yielded the same conclusion.

Drug repurposing, that is, the use of existing commercially available drugs as an alternative treatment for a novel disease, is often attempted, and is a measure that greatly reduces the cost of drug development [22,23]. The adverse effects of repurposed drugs have often been studied previously. However, the current World Health Organization guidelines on drugs for COVID-19 only strongly recommend two drugs (corticosteroids and interleukin-6 receptor blockers) as a treatment for patients with severe or critical COVID-19 [24]. Thus, verifying that existing commercial drugs are effective against COVID-19 by using a large-scale in-patient database is crucial.

We initially hypothesized that NM may improve patient outcomes, as experimental studies recently reported that NM inhibits the entry of SARS-CoV-2 into human epithelial cells [5,25], and several case reports have demonstrated the potential clinical benefits of NM [2,6,7,8]; however, we found no statistically significant association between NM and improvements in in-hospital mortality.

Our crude overall in-hospital mortality was 5.1%, which is lower than a large study previously conducted in Japan (in-hospital mortality rate, 11.6%), and the results showed lower mortality compared to other countries [26]. The reason may be few patients with severe COVID-19 were included (e.g., those admitted to the ICU and those requiring RRT or ECMO) in this study. We adjusted the severity by using propensity score matching; however, low in-hospital mortality and severity might affect our results.

Recently, a phase 2 open-label RCT in patients requiring nasal high-flow oxygen therapy and/or non-invasive mechanical ventilation with COVID-19 showed NM did not shorten the time to clinical improvement [9]. The result of the study supports our findings but further validation may be needed by larger RCTs.

This study has several limitations. First, it was a nonrandomized observational study. Therefore, the database did not include detailed clinical information on factors, such as symptoms, vital signs, and laboratory data. However, we included data from previous studies, which have associated advanced age, male sex, obesity, smoking, cardiovascular disease, diabetes mellitus, chronic lung disease, and cancer with increased COVID-19 mortality [27,28], and considered those factors in our study. In addition, key factors forming part of the Acute Physiologic and Chronic Health Evaluation scoring system and severity of pneumonia (e.g., A-DROP score) that are widely used for predicting mortality or adjusting severity were assessed in our study (such as the use of vasopressors instead of mean arterial pressure, mechanical ventilation instead of oxygenation, RRT instead of renal function, and JCS instead of mental status). Second, we could not obtain data on newly approved COVID-19 drugs, such as remdesivir and ciclesonide, as they had not yet been assigned drug codes. Third, we were unable to adjust for the number of patients with COVID-19 seen at each facility during the study period, which may have affected the outcomes. Lastly, we were unable to assess long-term outcomes post-hospital discharge.

## 5. Conclusions

Despite using the current largest available sample size that we know of to date, we found no statistically significant association between NM administration and improvements in the incidence of in-hospital mortality in patients with COVID-19. Thus, administering NM to improve the aforementioned outcomes may not yet be justified. Further studies elucidating the benefits and disadvantages of administering NM to patients with COVID-19 are warranted.

## Figures and Tables

**Figure 1 jcm-11-00116-f001:**
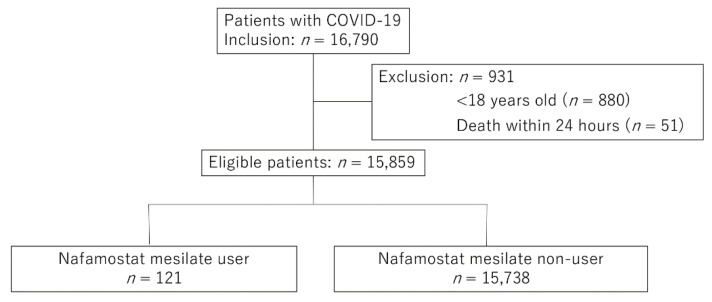
Flow chart of the patient selection process. COVID-19, coronavirus disease 2019.

**Figure 2 jcm-11-00116-f002:**
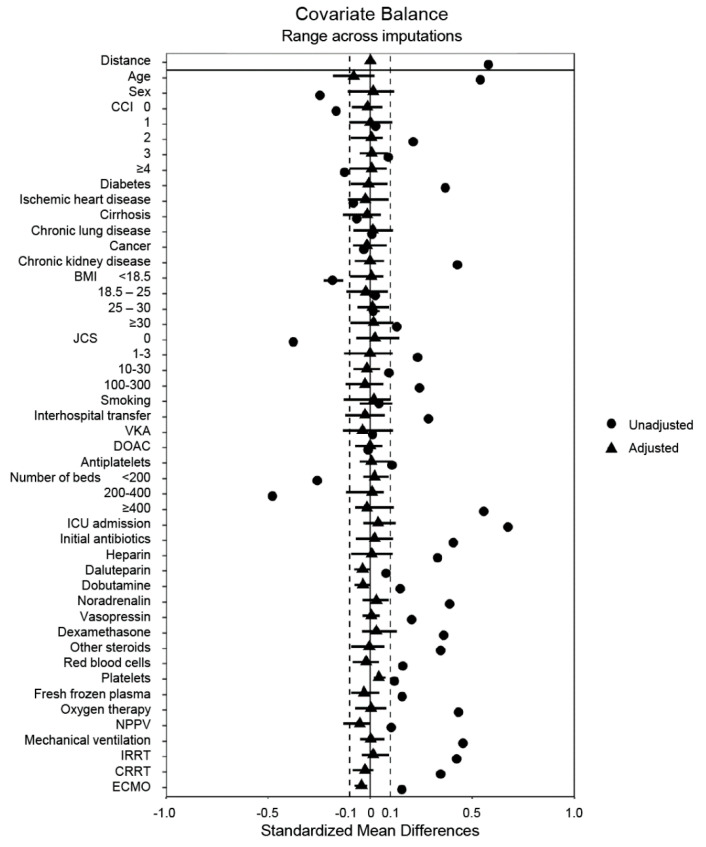
A summary plot of covariate balance before and after matching. BMI, body mass index; CCI, Charlson comorbidity index; CRRT, continuous renal replacement therapy; DOAC, direct oral anticoagulant; ECMO, extracorporeal membrane oxygenation; ICU, intensive care unit; IRRT, intermittent renal replacement therapy; JCS, Japan Coma Scale; NPPV, noninvasive positive-pressure ventilation; VKA, vitamin K antagonist.

**Table 1 jcm-11-00116-t001:** Unmatched patient characteristics categorized according to nafamostat mesylate use.

	Unmatched Group	
	Nafamostat Mesylate	Control
Number of patients	121	15,738
Age (years), mean ± SD ^1^	69.9 ± 15.0	61.8 ± 22.2
Male (%)	68.6	57.1
Charlson comorbidity index (%)		
0	43.8	52.1
1	9.9	9.1
2	25.6	16.4
3	8.3	5.8
≥4	12.4	16.6
Diabetes mellitus (%)	40.5	22.5
Ischemic heart disease (%)	5.8	7.7
Cirrhosis (%)	0.8	1.4
Chronic lung disease (%)	18.2	17.9
Cancer (%)	12.4	13.5
Chronic kidney disease (%)	24.8	6.4
Body mass index (%)		
<18.5	10.4	16.3
18.5–25.0	59.1	58.0
25–30	20.0	19.1
≥30	10.4	6.6
Smoking (%)	35.9	34.1
Japan Coma Scale (%)		
0 (clear)	66.1	83.9
1–3 (delirium)	22.3	12.7
10–30 (somnolence)	4.1	2.3
100–300 (coma)	7.4	1.1
VKA ^2^ (%)	1.7	1.5
DOAC ^3^ (%)	4.1	4.3
Antiplatelet (%)	9.1	6.1
Interhospital transfer (%)	16.5	5.9
Number of beds (%)		
<200	2.5	6.5
200–400	33.9	56.6
≥400	63.6	36.9
ICU ^4^ admission (%)	37.2	4.6

^1^ SD, standard deviation; ^2^ VKA, vitamin K antagonist; ^3^ DOAC, direct oral anticoagulant; ^4^ ICU, intensive care unit.

**Table 2 jcm-11-00116-t002:** Treatment within 2 days of admission in the unmatched cohort.

	Nafamostat Mesylate	Control
Initial antibiotics (%)	25.6	7.8
Heparin (%)	14.0	2.6
Daluteparin (%)	0.8	0.1
Dobutamine (%)	2.5	0.2
Noradrenalin (%)	14.9	1.0
Vasopressin (%)	4.1	0.1
Steroids		
Dexamethasone (%)	19.0	4.9
Other steroids (%)	19.8	6.1
Blood transfusion (%)		
Red blood cells (%)	4.1	0.9
Platelets (%)	1.7	0.1
Fresh frozen plasma (%)	2.5	0.0
Oxygen therapy (%)	47.9	26.3
NPPV ^1^ (%)	1.7	0.3
Mechanical ventilation (%)	19.8	1.7
IRRT ^2^ (%)	17.4	1.3
CRRT ^3^ (%)	10.7	0.1
ECMO ^4^ (%)	2.5	0.1

^1^ NPPV, noninvasive positive-pressure ventilation; ^2^ IRRT, intermittent renal replacement therapy; ^3^ CRRT, continuous renal replacement therapy; ^4^ ECMO, extracorporeal membrane oxygenation.

**Table 3 jcm-11-00116-t003:** Study outcomes after propensity score matching.

	Effect Estimate	*p*-Value
In-hospital mortality		
No nafamostat mesylate	1 (Reference)	
Nafamostat mesylate	1.27 (0.61–2.64)	0.52
Sensitivity analyses (in-hospital mortality)		
No nafamostat mesylate	1 (Reference)	
Nafamostat mesylate		
Exclusion of patients undergoing IRRT ^1^ or CRRT ^2^	1.03 (0.39–2.71)	0.94
Complete cases	1.32 (0.62–2.82)	0.46

^1^ IRRT, intermittent renal replacement therapy; ^2^ CRRT, continuous renal replacement therapy.

## Data Availability

The data presented in this study are available on request from the corresponding author.

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
