# Peer review of "Association between Nafamostat Mesylate and In-Hospital Mortality in Patients with Coronavirus Disease 2019: A Multicenter Observational Study"

_jcm, 2021, doi:10.3390/jcm11010116_

Round 1

Reviewer 1 Report

Very well presened data, in fact showing negetive result, but with a great practical soudness. Thank You. 

paper is absolutely porfect. Sufficient on statistical lever. Very important during pandemy. Recently there is some tendency to show only posiive results, but this time uselesness of one of medicals Nafamostat mesylate in practice is vey important and this paper should be prinstes asap.

Author Response

We deeply thank you for taking your time and effort to review our manuscript.

Reviewer 2 Report

Dear Authors,

I’ve read your manuscript. Your statistical analysis is very interesting, but very complex to understand. 

In effect the unmatched data reveal that NM was administered in a patients’ cohort deeply different from NM “non user” one (see tab. 1 and tab. 2). NM was given in severe acute respiratory failure (VAM 19.8% vs 1.7%; ECMO 2.5% vs 0.1%), renal failure (CRRT 10.7% vs 0.1%), haemodynamic instability (Noradrenaline 14.9% vs 1.0%). Moreover your research isn’t based on a data-base specifically designed for clinical purpose.

Moreover in line 54 you wrote that neither trials nor observational studies have demonstrated an association between NM and reduced mortality in Covid 19. Recently S. Zhuravel and coll (J. El. Clin Med 2021 Nov; 41: 101169 DOI: 10.1016/jelclinmed2021.101169) published a paper based on a phase 2 open-label, randomised controlled trial in that NM (4.8 mg/kg/day) administered in Covid 19 Pt requiring HFNC and/or NIV didn’t give any clinical improvement. So I think that you should insert this paper and its consclusion in your manuscript and in your references, especially because is according to you.

Finally I think that you should under lyne the need of future researches, based on multicenter trials regarding effect of NM in Covid 19 patients’ management, because, considering recently published papers, actually conflicting results are edited.

Best regards

Author Response

Comment 1:

I’ve read your manuscript. Your statistical analysis is very interesting, but very complex to understand. In effect the unmatched data reveal that NM was administered in a patients’ cohort deeply different from NM “non user” one (see tab. 1 and tab. 2). NM was given in severe acute respiratory failure (VAM 19.8% vs 1.7%; ECMO 2.5% vs 0.1%), renal failure (CRRT 10.7% vs 0.1%), haemodynamic instability (Noradrenaline 14.9% vs 1.0%). Moreover your research isn’t based on a data-base specifically designed for clinical purpose.

Response:

Thank you for carefully reviewing our manuscript and your thoughtful suggestions and insights.

Thank you for carefully reviewing our manuscript and your thoughtful suggestions and insights.

              As you suggested, we described that the unmatched data indicated that patients administered NM were more severe in the result rection; thus, we matched the baseline characteristics between the groups based on  propensity scores. Moreover, we conducted a sensitivity analysis excluding patients who had undergone intermittent or continuous RRT, which resulted in similar results to the main analysis.

In addition, as you are concerned, this study did not conduct primary data collection specidically for the study purpose, but used a secondary database of Japanese Diagnosis Procedure Combination fixed-payment reimbursement system in this study. However, under the COVID-19 pandemic, clinicians could not know whether NM was effective or not for COVID-19 in a clinical situation, whereas conducting a randomized clinical trial was extremely difficult and unfeasible in Japan. Thus, we believe that our study is the largest to date regarding the effectiveness of NM even if the study was based on secondary data.

Comment 2:

Moreover in line 54 you wrote that neither trials nor observational studies have demonstrated an association between NM and reduced mortality in Covid 19. Recently S. Zhuravel and coll (J. El. Clin Med 2021 Nov; 41: 101169 DOI: 10.1016/jelclinmed2021.101169) published a paper based on a phase 2 open-label, randomised controlled trial in that NM (4.8 mg/kg/day) administered in Covid 19 Pt requiring HFNC and/or NIV didn’t give any clinical improvement. So I think that you should insert this paper and its consclusion in your manuscript and in your references, especially because is according to you.

Finally I think that you should under lyne the need of future researches, based on multicenter trials regarding effect of NM in Covid 19 patients’ management, because, considering recently published papers, actually conflicting results are edited.

Response:

We appreciated your meaningful comment. We have added the result of the study you kindly suggested and inserted the reference. In the introduction, we have changed the sentence “neither clinical trials nor observational studies” to “neither clinical trials nor observational studies, except for a recent small a phase 2 open-label, randomised controlled trial (RCT) [9]” In discussion, we have added the sentence, “Recently, a phase 2 open-label RCT in patients requiring nasal high-flow oxygen therapy and/or non-invasive mechanical ventilation with COVID-19 showed NM did not shorten the time to clinical improvement [9]. The result of study supports our findings but further validation may be needed by larger RCTs.” with the reference.

Reviewer 3 Report

The manuscript is well written and presented. The Authors should be commended for their work and the way the study was conducted. I recommend only to unform the asmi in the abstract and the manuscript (Introduction session). I also recommend to present better quality images (i.e., Figure 2 shows not good quality).  

Author Response

Comment:

The manuscript is well written and presented. The Authors should be commended for their work and the way the study was conducted. I recommend only to unform the asmi in the abstract and the manuscript (Introduction session). I also recommend to present better quality images (i.e., Figure 2 shows not good quality).

Response:

We sincerely thank you for taking your time and effort necessary to review our manuscript. We have improved the quality of the Figures.